# Impact of De Novo Cholesterol Biosynthesis on the Initiation and Progression of Breast Cancer

**DOI:** 10.3390/biom14010064

**Published:** 2024-01-03

**Authors:** Danila Coradini

**Affiliations:** Laboratory of Medical Statistics and Biometry, “Giulio A. Maccacaro”, Department of Clinical Sciences and Community Health, University of Milan, Campus Cascina Rosa, 20133 Milan, Italy; danila.coradini@gmail.com

**Keywords:** cholesterol biosynthesis, Hippo signaling pathway, breast cancer initiation, breast cancer stem cells, in situ ductal carcinoma, invasive breast cancer, drug resistance, cholesterol-lowering treatment

## Abstract

Cholesterol (CHOL) is a multifaceted lipid molecule. It is an essential structural component of cell membranes, where it cooperates in regulating the intracellular trafficking and signaling pathways. Additionally, it serves as a precursor for vital biomolecules, including steroid hormones, isoprenoids, vitamin D, and bile acids. Although CHOL is normally uptaken from the bloodstream, cells can synthesize it de novo in response to an increased requirement due to physiological tissue remodeling or abnormal proliferation, such as in cancer. Cumulating evidence indicated that increased CHOL biosynthesis is a common feature of breast cancer and is associated with the neoplastic transformation of normal mammary epithelial cells. After an overview of the multiple biological activities of CHOL and its derivatives, this review will address the impact of de novo CHOL production on the promotion of breast cancer with a focus on mammary stem cells. The review will also discuss the effect of de novo CHOL production on in situ and invasive carcinoma and its impact on the response to adjuvant treatment. Finally, the review will discuss the present and future therapeutic strategies to normalize CHOL biosynthesis.

## 1. Introduction

Cholesterol (CHOL) is the most common sterol of vertebrates, and its homeostasis is a dynamic balance between absorption and de novo synthesis. While higher CHOL intake from food leads to a net decreased endogenous production, the body compensates for the poorly absorbed neutral cholesteryl esters by synthesizing de novo CHOL primarily in the liver [1]. Once synthesized, CHOL is transported through the bloodstream around the body, packaged within lipoprotein particles, mainly low-density lipoproteins (LDLs), which can carry up to 6000 fat molecules per particle [2].

Cellular CHOL homeostasis reflects this dynamic balance. Specific receptors (LDLRs), located on the plasma membrane, facilitate the uptake of LDLs, and once internalized, they are stored in endosomes, where they undergo lipase-dependent hydrolysis, making CHOL available for various cellular processes [3]. Through a feedback regulatory mechanism, this exogenously produced CHOL inhibits the de novo synthesis by repressing *HMGCR*, the gene coding for 3-hydroxy-3-methylglutaryl (HMG)-CoA reductase, the enzyme catalyzing the first rate-limiting step in the biosynthetic process (Figure 1).

The transcription of *HMGCR* is governed by sterol regulatory element-binding protein (SREBP) transcription factors, which are sensitive to CHOL concentration. In the presence of CHOL, SREBP precursors are retained in the membrane of the endoplasmic reticulum by a tight association with a multi-pass membrane protein called SREBP cleavage-activating protein (SCAP), which has a sterol-sensing domain to which CHOL can bind [4]. SCAP also forms a reversible ternary complex with another multi-pass membrane protein called the insulin-induced gene (INSIG), which promotes the ubiquitin-mediated degradation of HMG-CoA reductase by recruiting the membrane-associated ubiquitin ligase gp78 [5,6,7].

Conversely, under CHOL deprivation conditions, SCAP and INSIG no longer bind. INSIG is degraded, whereas SCAP undergoes a conformational change in the sterol-sensing domain that unmasks the endoplasmic reticulum export signal. The SCAP/SREBP complex is then packed into COPII-coated vesicles and targeted to the Golgi apparatus, where SREBP is sequentially cleaved by site-1 protease and site-2 protease (S1P and S2P) [5]. The N-terminal domain of SREBP resulting from cleavage can enter the nucleus, bind to sterol-responsive elements (SREs), and activate the transcription of both the gene coding for the LDL receptor to enhance the intracellular CHOL uptake and those coding for HMG-CoA reductase and squalene epoxidase (SQLE), the enzyme that catalyzes the irreversible commitment to CHOL, to increase the endogenous production.

Several studies have shown that CHOL metabolism is often reprogrammed in cancer cells, and it may play a crucial role in the development and progression of several types of cancer, such as colon, rectal, prostatic, testicular, and breast cancer [8,9,10,11,12,13]. This review will examine the contribution of de novo CHOL biosynthesis to the pathophysiology of breast cancer. After an overview of the multiple biological activities of CHOL and its major derivatives (Section 2), this review will address the impact of de novo CHOL production on the promotion of breast cancer, focusing on mammary stem cells (Section 3.1). Then, the review will discuss the effect of de novo CHOL production on in situ (Section 3.2) and invasive carcinoma (Section 3.3) and its impact on the response to the adjuvant treatment (Section 4). Finally, the review will discuss the current and future therapeutic strategies to normalize CHOL biosynthesis (Section 5).

## 2. Cholesterol Biology

### 2.1. Cholesterol and Membrane Dynamics

CHOL makes up about 30% of cell membranes. Due to its unique ability to keep the membrane in a state of microfluidity over the range of physiological temperatures, CHOL is essential for cellular homeostasis. It achieves this by regulating crucial processes such as membrane reorganization, cell trafficking, and signal transduction [14,15]. This capability is due to the quasi-rigid tetracyclic structure of CHOL (Figure 1), which enables it to sit perpendicular to the membrane surface. The hydroxyl (-OH) group interacts with the water molecules surrounding the membrane while the bulky steroid and the hydrocarbon chain branch to the membrane alongside the nonpolar fatty acid chain of the other lipids, thus increasing membrane packing. In particular, CHOL organizes membrane compartmentalization by creating specific ordered domains called rafts. These structures float freely in the disordered fluid phase of the rest of the membrane and can cluster to form large rafts [16,17]. Rafts provide a privileged docking site for membrane-associated receptors, adhesion molecules, ion channels, and regulatory molecules, which are involved in essential cellular functions such as signal transduction, intracellular trafficking, and adhesion to the extracellular matrix [18,19,20,21].

Recent evidence suggests that cancer cells contain more lipid rafts than their non-tumorigenic counterparts and that the disruption of their integrity is associated with the activation of many signaling processes involved in cancer development and progression [22,23].

### 2.2. Cholesterol Direct Activity

In addition to the membrane-associated activities, CHOL can also act as an endogenous agonist to regulate some signaling pathways. The most well-studied of these signaling pathways is the estrogen-related receptor α (ERRα) pathway [24]. ERRα is an orphan nuclear receptor that senses energy levels and promotes the expression of the genes involved in metabolic processes like β-oxidation, oxidative phosphorylation, the tricarboxylic acid cycle, and glycolysis when energy demand is high [25]. Like many other nuclear receptors, ERRα transcriptional activity depends on the presence of coregulators, particularly peroxisome proliferator-activated receptor-γ (PPARγ) coactivator 1 (PGC-1) [26].

Research has shown that CHOL can control ERRα activity by binding to its ligand-binding domain through its -OH group, enhancing the recruitment of the PGC-1 coactivator and increasing the transcription of the genes required for maintaining energy homeostasis [27].

Assuming that metabolic adaptations are a hallmark of cancer cells [28] and the importance of ERRα/PGC-1 axis in controlling cellular adaptation to energy demand, it is not surprising that in breast cancer, the hyperactivation of ERRα, due to increased availability of CHOL leads to metabolic reprogramming aimed at promoting and maintaining energy-consuming processes such as cell proliferation and migration [29,30] and providing a mechanistic explanation for the increased breast cancer risk associated with high dietary CHOL [31].

Noteworthily, recent studies also suggest that the CHOL-mediated ERRα activation can cause metabolic switching, leading to the propagation of cancer stem-like cells (CSCs) [32], triggering the epithelium–mesenchymal transition (EMT), the release of pro-inflammatory factors that can modify the tumor microenvironment [31], and the onset of drug resistance [33], as supported by the reduced ERRα transactivation observed after the treatment with statins [27].

### 2.3. Cholesterol Derivatives

In addition to the direct activity on membrane dynamics and cellular signaling transduction, CHOL serves as a parent compound for several biomolecules, such as bile acids, oxysterols, and steroid hormones.

The synthesis of bile acids from CHOL occurs primarily in the liver and generates several distinct bile acids. These bile acids act as solvents facilitating the absorption of dietary fats and oils and also activate G protein-coupled receptors (GPCR), thus regulating the expression of genes not only involved in bile acid metabolism but also in glucose homeostasis, lipid and lipoprotein metabolism, energy expenditure, intestinal motility, inflammation, and liver regeneration [34]. The conversion of CHOL to bile acids promotes CHOL efflux via the liver X receptor (LXR) and suppresses de novo CHOL production [35], thus maintaining CHOL homeostasis.

Bile acid synthesis can occur via two pathways. The classic (neutral) pathway is activated by the hepatic enzyme cholesterol 7α-hydroxylase (CYP7A1) that converts CHOL to 7α-hydroxycholesterol (7α-OH-Chol). Triggered by the mitochondrial enzyme sterol 27-hydroxylase (CYP27A1), the acidic pathway forms 27-hydroxycholesterol (27-OH-Chol), the most abundant oxysterol derivative.

Studies have shown that 27-OH-Chol can bind the estrogen receptor (ER) and acts as an endogenous selective estrogen receptor modulator (SERM) with detrimental effects on breast cancer pathogenesis, acting as a partial agonist and promoting the growth of ER-positive cells [36,37,38,39]. The concentration of 27-OH-Chol in ER-positive breast cancer was up to six-fold higher than the adjacent normal tissue. Such an increased concentration was negatively associated with disease-free survival, mainly in post-menopausal women, probably because of its agonistic action [40,41]. Furthermore, 27-OH-Chol may indirectly promote tumor growth and metastasis spread by decreasing immune surveillance [42].

Steroid hormones are, likely, primarily the CHOL derivatives. They are produced through a multistep process termed steroidogenesis [43], and they play an essential role in many physiologic processes, including energy homeostasis, metabolism control, inflammation, immune functions, salt and water balance, and the development of sex-specific characters [44,45].

However, steroid hormones also contribute to several metabolic disorders, primarily obesity [46,47], and to the development of various hormone-sensitive tumors, including breast cancer. Sex steroid hormones and their receptors are recognized as playing an essential role in breast cancer development and progression, as supported by the evidence that approximately 70% of all breast cancers are dependent upon steroid hormones for growth and constitutively express the corresponding receptors [48] through which these hormones act directly on cancer cells and indirectly on the cancer microenvironment [49,50,51].

Steroid hormones can act through two different mechanisms depending on the type of bound receptor: nuclear or membrane-associated [52]. The genomic or nuclear mechanism is slow and requires the free estradiol to pass the cell membrane. Once in the cytoplasm, estradiol binds to its specific receptor (ER) and forms a functional complex that moves to the nucleus. Here, the complex binds to specific DNA sequences and regulates the transcription of different genes. Conversely, the non-genomic or extra-nuclear mechanism is much faster because it acts through membrane-associated receptors, such as GPCR and mER, triggering diverse intracellular signaling cascades [53,54].

## 3. Cholesterol Biosynthesis and Breast Cancer Initiation and Development

Elevated levels of circulating CHOL due to CHOL homeostasis dysregulation can cause severe metabolic dysfunctions. High serum levels of CHOL are a well-known risk factor for heart disease [55], and cumulating evidence suggests that hypercholesterolemia may also be associated with the risk of developing several types of tumors, including breast cancer [56,57,58]. However, observational studies on the relationship between circulating CHOL and breast cancer risk have provided inconclusive or sometimes contradictory results. While some systematic reviews and meta-analyses suggest that dietary CHOL intake and serum CHOL levels are associated with breast cancer risk [59,60], other prospective studies have found no significant or inverse association [61,62,63]. Confounding variables such as obesity, menopausal status, tumor subtype, and ER status may be the reason for these conflicting results.

Conversely, preclinical studies have provided compelling evidence that tumors show a CHOL concentration higher than normal tissue, indicating a significant change in CHOL homeostasis [8]. This finding agrees with the increased requirement for CHOL of transformed cells to sustain their abnormal proliferation. They satisfy this requirement by increasing the uptake of exogenous CHOL through the overexpression of LDL receptors and the endogenous production through de novo biosynthesis. Recent findings have also shown that the dysregulation of the mevalonate pathway, the core of the biosynthetic process (Figure 1), results in the overproduction of isoprenoids [64] that play an essential role in the posttranslational modification (prenylation) of several proteins, including Rho GTPases (Rac, RhoA, Cdc42). These proteins act as molecular switches in various cell processes and signaling pathways, including the Hippo signaling pathway.

### 3.1. Cholesterol Biosynthesis and the Hippo Signaling Pathway

First discovered in *Drosophila*, the evolutionarily conserved Hippo signaling pathway limits organ size by controlling tissue growth and preventing the uncontrolled proliferation of progenitor cells [65]. The core of the Hippo pathway consists of two highly conserved serine/threonine kinases called serine/threonine kinase 3/4 (STK3/4), also known as mammalian Ste20-like kinase 1/2 (MST1/2), and large tumor suppressor 1/2 (LATS1/2). When cells stop growth in response to cell contact inhibition, these kinases phosphorylate the Yes-associated protein 1 (YAP)/transcriptional coactivator with PDZ-binding motif (TAZ) coactivators [66], preventing them from entering the nucleus and promoting their degradation.

As schematically depicted in Figure 2, the signaling cascade starts when MTS1/2 associates with the scaffold protein Salvador family WW domain-containing protein 1 (SAV1) to phosphorylate and activate LATS1/2. Once activated, LATS1/2 associates with the regulatory MOB kinase activator 1/2 (MOB1/2) and phosphorylates the YAP/TAZ coactivator on multiple sites, including Ser168, which functions as the binding site for 14-3-3 protein. The latter sequesters YAP/TAZ in the cytoplasm and primes for its rapid proteasomal degradation, thereby inhibiting their migration into the nucleus [67]. Conversely, when YAP/TAZ coactivators are not phosphorylated, they can enter the nucleus, associate with TEA domain transcription factor 1/4 (TEAD1/4), and regulate the expression of several genes involved in cell proliferation promotion and inhibition of apoptosis.

Cumulating evidence indicated that Rho GTPases play a crucial role in promoting the nuclear translocation of YAP/TAZ. Rho GTPases are usually found free in the cytoplasm. However, when structurally modified by the enzymatic activity of the geranylgeranyl pyrophosphate produced by the mevalonate cascade, they anchor to cell membranes and act as a transducer of the changes occurring in the cytoskeleton dynamics. In response to growth factors, Rho GTPases coordinate the rearrangement of the actin stress fibers, which, in turn, sequester angiomotin, an inhibitor of YAP/TAZ, and promote the nuclear translocation of the coactivators [68,69]. Previously considered independent, the Rho GTPase-mediated activation of YAP/TAZ is now emerging as linked to the Hippo pathway because of the direct inhibition of LATS1/2 kinase [70].

Given the role played by the Hippo signaling pathway in controlling the growth of normal cells, it is not surprising that the disruption of this pathway leads to the constitutive activation of YAP/TAZ, which in turn promotes uncontrolled proliferation and tumorigenesis [66,71], and the finding that common solid tumors, including breast cancer, show a high expression of YAP protein [72].

### 3.2. Cholesterol Biosynthesis, Mammary Stem Cells, and Breast Cancer Stem Cells

The mammary gland is a highly dynamic organ that undergoes profound changes throughout the woman’s reproductive life. Starting at puberty and until menopause, these changes are more noticeable during pregnancy and lactation when the glandular tissue proliferates, undergoes complete differentiation, and secretes milk due to the influence of sex and lactogenic hormones.

The functional structure of the human mammary gland consists of 6–12 independent ductal–lobular systems drained by collecting ducts that converge at the nipple (Figure 3). Each ductal-lobular system contains several terminal ductal–lobular units (TDLUs) that represent the secretory unit of the mammary gland. TDLUs contain lobules or alveoli drained by a duct. The alveoli are composed of lobular epithelial cells, while the ducts are composed of epithelial cells (facing the lumen) and myoepithelial cells that form the basal layer. TDLUs are surrounded by stroma, which contains different kinds of cells such as adipocytes, endothelial cells, immune cells, and fibroblasts, all of which are embedded in the extracellular matrix.

Epithelial and myoepithelial cells derive from common ancestors known as multipotent mammary epithelial stem cells (MaSCs), which have the unique capacity to self-renew and to generate the lineage-restricted progenitors of both lobular and ductal epithelial cells as well as myoepithelial cells [73,74,75]. These progenitors, located in distinct areas of the developing mammary known as niches, proliferate extensively and differentiate to form TDLUs under the control of ovarian hormones [76,77].

Since the discovery of MaSC, significant progress has been made in identifying the pathways that control the self-renewal, the lineage commitment, and the differentiation of stem and progenitor cells during the development of the normal mammary gland and their potential involvement in cancer initiation [78,79]. There is now cumulating evidence that the signaling pathways that regulate the physiological turnover of the mammary tissue are often disrupted in breast cancer, suggesting that stem and progenitor cells, due to critical gene mutations and epigenetic events, can give rise to a small population of highly tumorigenic cells known as breast cancer stem cells (BCSCs) [80,81,82,83].

Recent preclinical studies have found that CSC-enriched breast cancer cell lines have an increased expression of the enzymes involved in the de novo synthesis of CHOL compared to control cells [84]. This finding suggests that CHOL biosynthesis plays a crucial role in BCSC propagation and tumor development, especially considering the link between estradiol, the most potent bioactive CHOL derivative, and breast cancer growth. Indeed, estradiol promotes and fuels the proliferation of estrogen-dependent progenitors and contributes to the development of ER-positive tumors that account for about 70% of breast cancer diagnoses.

Additionally, studies have shown that the mevalonate metabolic pathway is remarkably hyperactivated in CSC-enriched breast cancer cell lines. However, this hyperactivation was reduced by treatment with statins, leading to decreased isoprenoid production and less availability of anchorage sites to the membrane for the Rho GTPases [85]. This finding suggests that the dysregulation of CHOL biosynthesis, which leads to the over-production of isoprenoids, may modify the physiological turnover of stem and progenitor cells, promoting their transformation and proliferation.

Interestingly, the downregulation of the mevalonate precursor enzyme, HMG-CoA synthase (HMGCS1), through silencing the expression of the corresponding gene, resulted in a significant decrease in BCSC fraction. This finding proposes HMGCS1 as a potential gatekeeper for the mevalonate pathway metabolism, whose dysregulation enhances cancer stem cell enrichment [86]. Blocking the expansion of BCSCs through inhibition of the *HMGCS1* gene could be a more efficient approach to control the over-production of isoprenoids compared to standard doses of statins, thereby reducing the proliferation of progenitor cells and cancer development.

Another promising approach to prevent the expansion of BCSCs by regulating CHOL biosynthesis involves microRNAs (miRNAs). They are small (~18–25 nucleotides) non-encoding RNAs that can modulate the expression of the genes involved in the pathway [87]. A recent study using bioinformatics analysis [88] has revealed that hsa-miR-34a-5p and hsa-miR-373-3p can affect the expression of several genes related to CHOL, including *INSIG2.* This gene codes for insulin-induced gene 2, an endoplasmic reticulum protein that acts as a negative regulator of CHOL biosynthesis by retaining the SCAP-SREBP complex in the endoplasmic reticulum. As a result, SREBP processing was blocked, and the transcription of the target genes, including *HMGCS1,* was inhibited.

### 3.3. Cholesterol Biosynthesis and Ductal Carcinoma In Situ

Ductal carcinoma in situ (DCIS) is a non-invasive form of breast cancer whose incidence has risen markedly since the early 1980s due to the widespread use of mammography screening. According to the American Cancer Society, in 2022, DCIS accounted for 15% of all new mammographically detected breast cancers, 82% of which occurred in women aged 50 years and older [89]. Although DCIS is considered a non-obligate precursor to invasive breast cancer (IBC), if left untreated, about 50% of cases progress to IBC, even several decades after diagnosis. A 30-year follow-up study showed that 36% of untreated low-grade DCIS progressed to IBC [90].

While many efforts have been made to elucidate the molecular changes associated with DCIS progression [91,92,93], less attention has been given to the molecular and metabolic changes that lead to the transformation of the normal mammary epithelium in DCIS. Identifying the early molecular events that trigger and drive this transformation is essential to prevent DCIS with appropriate and effective treatments.

In a recent study, we investigated the expression of some genes involved in CHOL biosynthesis in a set of patient-matched samples of DCIS and corresponding histologically normal (HN) epithelium [94]. We found that *HMGCR*, which encodes the enzyme that controls the rate-limiting step in CHOL biosynthesis, was more expressed in the epithelial compartment of DCIS than in the corresponding normal epithelium. This finding agrees with the clinical evidence that the HMG-CoA reductase (evaluated as immunohistochemical cytoplasmic staining) is moderately/strongly expressed in about 70% of the assessed DCIS samples [95]. Similarly, some crucial downstream genes involved in CHOL (*FDFT1*, *SQLE*, and *LSS*) and isoprenoid production (*GGPS1*) were more expressed in DCIS.

We also found that *HMGCS1*, the gene coding for the enzyme upstream to the mevalonate pathway, was more expressed in DCIS than in the corresponding normal tissue. This result supports the hypothesis that DCIS may originate from transformed progenitor cells, where the overexpression of HMG-CoA synthase plays a crucial role in dysregulating the biosynthetic process.

Apart from the differential expression, the genes involved in CHOL biosynthesis changed their interrelationship dramatically in DCIS. Thus, for example, the positive association between *HMGCR* and the downstream *GGPS1*, observed in normal tissue, disappeared in DCIS, and the positive association between *GGPS1* and *SQLE*, which regulate, respectively, the production of isoprenoids and the irreversible commitment to CHOL [96], switched to a negative one. The result suggests that, due to the dysregulation of the biosynthetic process, CHOL production was overcome by that of isoprenoids.

Collectively, these results support the hypothesis of a link between the dysregulation of the mevalonate pathway and the transcriptional activity of the YAP/TAZ coactivator through the increased production of the isoprenoids required for the prenylation of small GTPases, followed by GTPases anchoring to the cell membrane and activation, inhibition of the Hippo signaling pathway and promotion of YAP/TAZ coactivator nuclear translocation. Moreover, these results substantiate the use of CHOL-lowering drugs to restore the correct equilibrium in the production of isoprenoids and CHOL, thereby blocking cell proliferation and preventing DCIS progression to a more aggressive form.

### 3.4. Cholesterol Biosynthesis and Invasive Breast Cancer

Considering the natural history of breast cancer, it is not surprising that CHOL biosynthesis is dysregulated in IBC, where CHOL and its derivatives contribute to cancer cell proliferation and dissemination [97]. Preclinical studies have shown that CHOL not only triggers metabolic switching through the activation of the ERRα pathway (see Section 2.2) but also contributes to cancer cell invasion by increasing tumor sphere formation [98] and the number of the cholesterol-enriched rafts [99], where the membrane-associated receptors for growth factors, adhesion molecules, and immunoregulatory molecules [100] are concentrated.

In a recent study aimed at exploring the molecular basis of CHOL biosynthesis dysregulation in paired samples of IBC and adjacent HN tissue [101], it was found that the majority of the genes coding for the enzymes involved in CHOL biosynthesis were more expressed in tumors than the corresponding HN tissues. In particular, the genes differentially expressed code for the enzymes involved in rate-limiting or commitment steps of the biosynthetic process: HMG-CoA synthase (coded by *HMGCS1*), is recognized as the gatekeeper of the pathway, HMG-CoA reductase (coded by *HMGCR*), is known as the first rate-limiting enzyme, squalene monooxygenase (coded by *SQLE*), is the first committed step of cholesterol biosynthesis, and *NSDHL* that codes for the NAD(P)-dependent steroid dehydrogenase-like, has recently been associated with an unfavorable prognosis and metastatic spread [102].

The study also found that the expression level of *HMGCS1*, *HMGCR*, *SQLE*, and *NSDHL* progressively increased with tumor grade and correlated positively with the expression of *MKI67*, the gene coding for the proliferation marker Ki67. The association between the dysregulated CHOL biosynthesis and tumor aggressiveness was confirmed by the inverse association between the tumor expression level of *HMGCR* or *NSDHL* and the patient’s relapse-free survival. Even after taking into account tumor grade and MKI67 expression level as confounding covariates, patients with high expression levels of both genes had a significantly higher risk of relapse than those with low expression levels.

The finding that the expression profile of the genes involved in de novo *CHOL* biosynthesis progressively changed with tumor grade is particularly significant, considering that 40–60% of the diagnosed breast cancers fall in the heterogeneous class of G2 tumors which are characterized by a highly variable morphology, unpredictable risk of distant metastasis recurrence, [103] and not always adequate treatment [104]. Therefore, evaluating the expression profile of this panel of genes could significantly improve the classification of G2 tumors and help select the most effective therapeutic approach to improve the patient’s prognosis.

## 4. Cholesterol Biosynthesis and Drug Resistance

Acquired resistance to adjuvant chemotherapy or endocrine treatment is a significant drawback that increases the risk of relapse and mortality for breast cancer patients.

Cumulating evidence suggests that changes in body CHOL homeostasis and tumor CHOL metabolism play a role in developing and maintaining drug resistance and disease progression [105,106] and that CHOL-depleting or CHOL-lowering drugs may help overcome the resistance.

Preclinical studies have shown that in ER-positive breast cancer cells, the addition of acetyl plumbagin, a CHOL depletor, overcomes the acquired resistance to tamoxifen [107] by deactivating the PI3K/Akt/mTOR signaling pathway, which is the major endocrine resistance pathway [108], and that CHOL-lowering statins improved the apoptotic effect of doxorubicin or docetaxel and the sensitivity to cisplatin in triple-negative breast cancer cells [109,110].

These findings have been confirmed by clinical evidence from studies such as the Breast International Group (BIG) 1-98 study, which enrolled postmenopausal women with early stage, ER-positive IBC [111]. The study demonstrated that CHOL-lowering statins during adjuvant endocrine therapy with tamoxifen (antiestrogen) or letrozole (aromatase inhibitor), improved the disease-free-survival and the distant recurrence-free interval. Similar findings were observed in patients with metastatic triple-negative breast cancer and treated with chemotherapy. The use of lipophilic statins improved the prognosis of patients by prolonging tumor dormancy and reducing of about 50% the development of liver metastases [112].

To better understand the role played by de novo CHOL biosynthesis in the development of acquired resistance, a recent study compared the expression of the genes involved in the biosynthetic process and their association with the genes involved in estrogen production, the Hippo signaling pathway and cell cycle control, in a series of patients with ER-positive tumor responsive and nonresponsive to letrozole [113]. The study found that in responsive tumors, letrozole significantly decreased the expression of *HMGCS1*, *HMGCR*, *FDPS*, and *SQLE*. Conversely, in nonresponsive tumors, these genes were unaffected by letrozole but associated with some genes involved in estrogen production (*CYP19A1*, *HSD17B2*, and *SULT1A*, respectively, coding for aromatase, the enzyme that converts androgens in bioactive estrogens, hydroxysteroid 17-beta dehydrogenase 2, the enzyme that converts the estradiol to the less active estrone, and sulfotransferase family 1A member 1, the enzyme that neutralizes the estradiol by converting it into estradiol-sulfate), cell cycle control (*CDK4* and *CDK6*, coding the two cyclin-dependent kinases that drive the G1-phase of the cell cycle and proved to be essential for the physiologic mammary development and proliferation of ER-positive breast cancers [114,115])*,* and Hippo pathway (*YAP1*). Overall, the findings support the notion that the dysregulation of the mevalonate pathway may contribute to the resistance to letrozole and cancer progression, probably through the overproduction of isoprenoids, and suggest the use of statins to reduce the production of bioactive estrogens and inhibit the YAP/TAZ-dependent transcriptional activity.

## 5. Conclusions and Perspectives

There is strong evidence from basic research and animal studies showing that increased de novo CHOL biosynthesis concurs with the initiation, development, and progression of breast cancer, promoting and maintaining cell proliferation and enhancing acquired resistance to chemotherapy or endocrine therapy. CHOL directly activates the ERRα signaling pathway and, indirectly, enhances and sustains breast cancer cell proliferation by over-production of isoprenoids, anchoring to the cell membrane and activating small GTPases, which in turn, inhibit the Hippo signaling pathway and promote YAP/TAZ coactivator translocation and accumulation of into the nucleus. Moreover, CHOL fuels cell proliferation by sex steroid hormones in ER-positive tumors, and its major oxysterol derivative (27-OH-Chol) enhances the dissemination of breast cancer cells through an agonistic action [40,41] and decreases immune surveillance [42].

Furthermore, the dysregulation of CHOL biosynthesis is observed in MaSCs, promoting the development of abnormally proliferating progenitor cells and the onset of benign tumors that may evolve in breast cancer, an event of paramount importance given the increasing incidence rate pointed out by the most recent breast cancer statistics [89] and likely due to the widespread use of mammographic screening.

Restoring a correct CHOL biosynthesis and a balanced production of isoprenoids and steroid hormones, therefore, is considered a promising strategy to prevent the metastatic spread of frank breast cancer due to the acquired resistance to adjuvant therapy, the progression of in situ to invasive cancer and tumor initiation from abnormally proliferating progenitors.

Researchers have considered several strategies to achieve this objective, the most feasible of which is the inhibition of HMG-CoA reductase activity by safe and low-cost statins or interfering selectively with farnesyl pyrophosphate synthase and preventing the synthesis of farnesyl pyrophosphate and geranylgeranyl pyrophosphate by bisphosphonates [116,117,118] (Figure 4). Consequently, statins are used to reduce breast cancer recurrence and mortality [119,120], and bisphosphonates are used to treat patients with osteolytic bone metastases [121].

To hamper statin resistance [122] and the accumulation of prenylated proteins in cells, the combination of statins and bisphosphonates has been proposed based on experimental [123] and observational findings [124,125], even though no clinical trials have yet been carried out to confirm this hypothesis.

Over the last years, researchers have repurposed some drugs, other than statins, as anticancer agents. One of these drugs is metformin, an antidiabetic agent used as a first-line treatment for type 2 diabetes. Preclinical studies have shown that metformin has anticancer properties and can lower cellular CHOL content in breast cancer cells. Sharma et al. [126] have shown that treating ER-negative breast cancer cells with metformin led to a significant reduction in CHOL content and inhibition of *HMGCR*, *LDLR*, and *SREBP1* genes, which play a crucial role in regulating CHOL homeostasis.

In addition to the repositioning of drugs that are already in use for other diseases, basic research is exploring alternative therapeutic strategies to block de novo CHOL synthesis more efficiently by targeting enzymes that are distal to HMG-CoA reductase, such as squalene synthase (encoded by SQLE gene) and lanosterol synthase (encoded by LSS gene), using nonstatin inhibitors. Several classes of potential squalene synthase inhibitors have been investigated, including allylamines (FR194738, NB-598), squalene analogs (trisnorsqualene alcohol), and some natural compounds of selenium and tellurium. Among these candidates, lapaquistat acetate (TAK-475), a benzoxazepine derivative, is the only one that has been evaluated in advanced clinical trials, either alone or in combination with statin [127]. However, due to toxic effects such as the abnormal elevation of several liver enzyme levels, any further clinical development has been stopped.

Several inhibitors of lanosterol synthase have been identified, among which RO48-8071 is the most potent [128]. Preclinical studies have shown that RO48-8071 efficiently reduces ER-positive human breast cancer cell viability and prevents tumor growth when administered to mice with BT-474 tumor xenografts without any apparent toxicity or adverse effect on the viability of normal human mammary cells [129]. However, despite these promising results, no clinical study has been yet carried out.

Finally, apart from the “conventional” pharmacological approach, researchers are also examining innovative “functional” strategies based on the silencing technique and non-coding RNA to control the biosynthetic process by modulating the expression of pivotal genes such as *HMGCS1* and *INSIG2* [10,130], opening very promising perspectives.

## Figures and Tables

**Figure 1 biomolecules-14-00064-f001:**
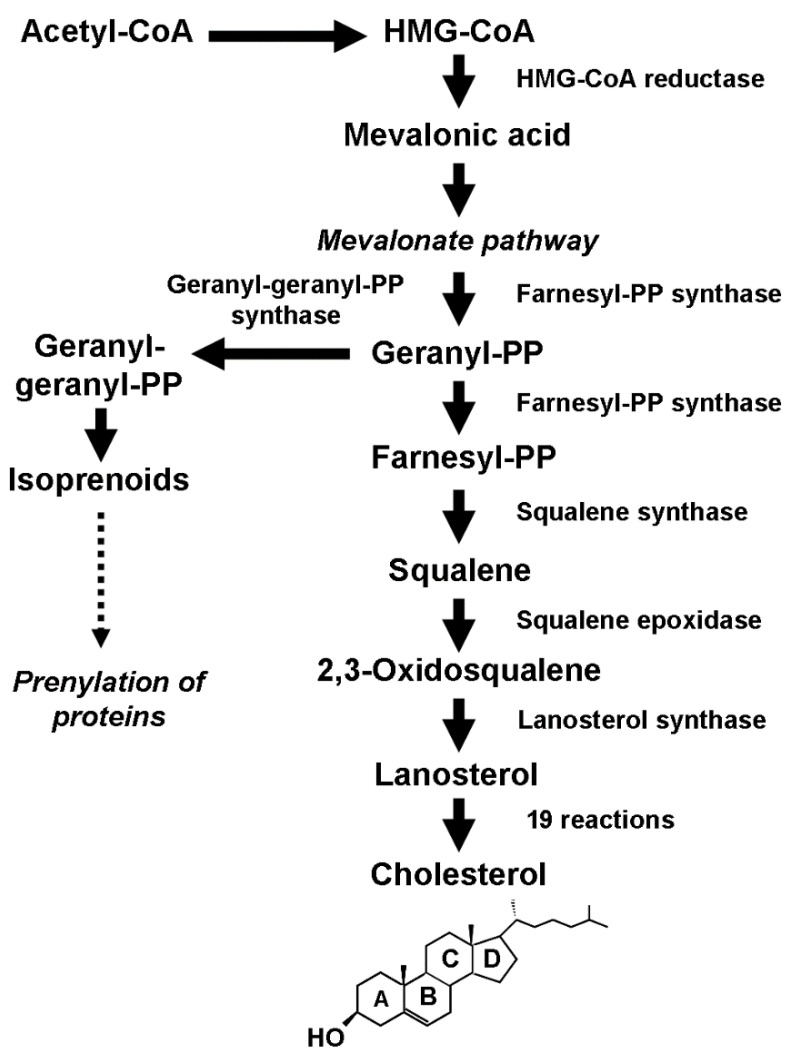
Schematic description of the biosynthetic process that leads to the production of cholesterol and isoprenoids.

**Figure 2 biomolecules-14-00064-f002:**
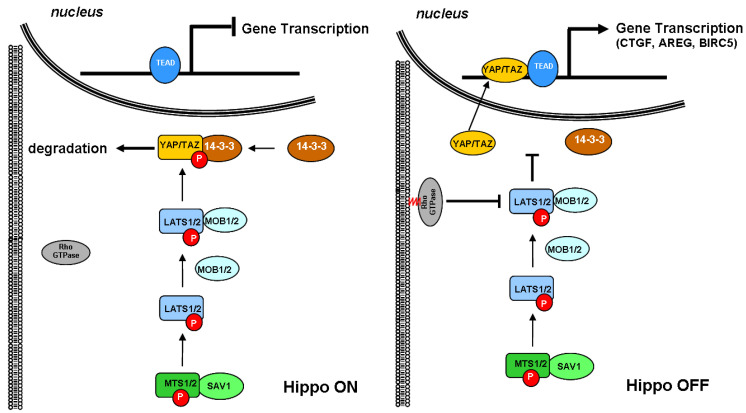
Schematic depiction of the interaction between small GTPase and Hippo signaling pathways. The core of the Hippo pathway consists of a kinase cascade where STK3, in complex with its regulatory protein SAV1, phosphorylates and activates LATS1. After forming a complex with the regulatory protein MOB1A, LATS1 phosphorylates and inactivates YAP. Phosphorylation of YAP by LATS1 inhibits its translocation into the nucleus, the binding to TEAD, and the transcription of several genes involved in cell proliferation, apoptosis, and migration. TEAD, TEA domain transcription factor; YAP, Yes-associated protein; TAZ, transcriptional coactivator with PDZ-binding motif; LATS1, large tumor suppressor kinase 1; MOB1A, MOB kinase activator 1A; STK3, serine/threonine kinase 3; SAV1, Salvador family WW domain-containing protein 1.

**Figure 3 biomolecules-14-00064-f003:**
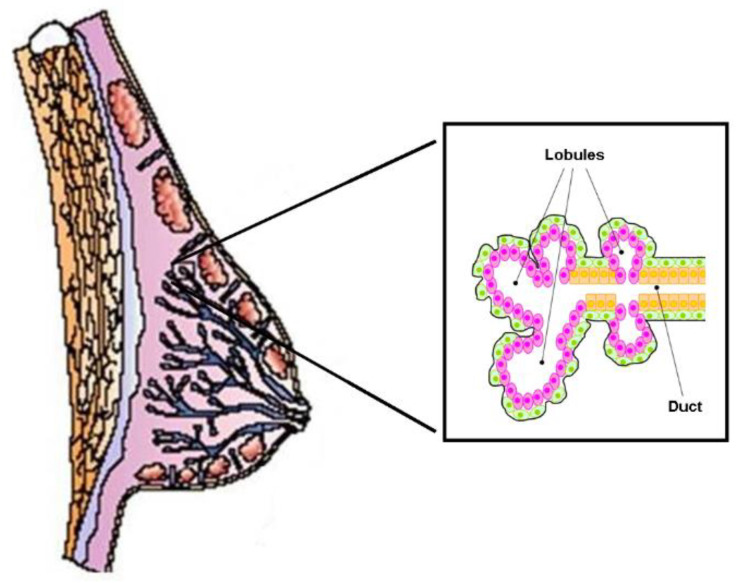
Schematic structure of the normal mammary gland. It consists of independent duct lobular systems drained by collecting ducts that converge at the nipple. The inset shows the cellular components of the terminal ductal–lobular unit (TDLU), the basic secretory unit of the mammary gland. Lobular (pink) and ductal (orange) epithelial cells are separated from the surrounding stroma by myoepithelial elements (in green).

**Figure 4 biomolecules-14-00064-f004:**
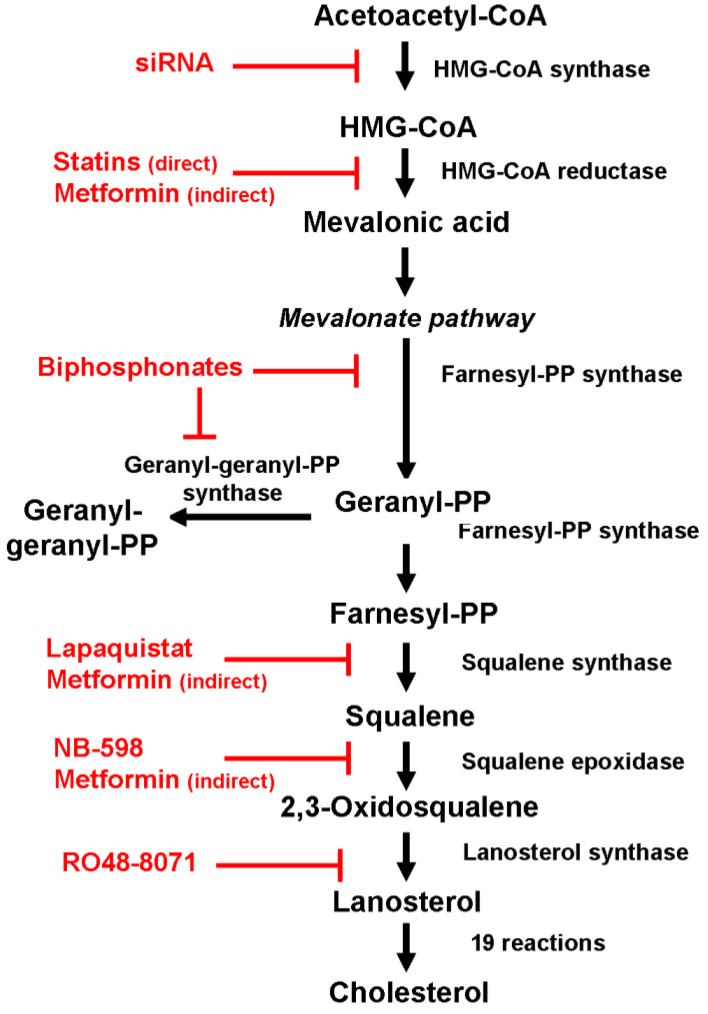
Repositioning drugs and innovative approaches to inhibit the activity of the enzymes involved in cholesterol biosynthesis.

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
