# Peer review of "Impact of De Novo Cholesterol Biosynthesis on the Initiation and Progression of Breast Cancer"

_biomolecules, 2024, doi:10.3390/biom14010064_

Round 1

Reviewer 1 Report

Comments and Suggestions for Authors

Cholesterol (CHOL) biosynthesis is an important topic to consider in breast cancer as CHOL is a precursor for estrogen and other steroid hormones that play a critical role in ER+ breast cancer. In addition, hypercholesterolemia is a risk factor for endocrine therapy resistance and recurrence in breast cancer. Thus, a review of CHOL and the CHOL synthesis pathway as they relate to breast cancer is relevant and impactful.

First, the author’s review of the basic cholesterol biosynthesis pathway (Fig 1) in Section 1 is informative and serves as a guide throughout the sections in the manuscript. Section 2 is also informative on the specific mechanisms of CHOL directly binding to ERRa and of CHOL derivatives as a partial agonist of ER in breast cancer.

In Section 3, the author attributes Rho geranylation leading to oncogenic YAP/TAZ activation is due to activation of “de novo CHOL synthesis.” However, mevalonate pathway activation (resulting in isoprenoids production) cannot be used synonymously with CHOL synthesis pathway. If anything, LATS as part of the Hippo pathway inhibits the SREBP transcription factor, thereby inhibiting CHOL synthesis. It feels that the author overly simplified Hippo/CHOL pathway interactions.

As for the CHOL synthesis pathway playing a role in stem cell, initiation, progression, the studies cited in this paper are mostly correlative studies (e.g. increased CHOL enzymes/genes), not mechanistic evidence. Moreover, many of the studies that the author cites are about mevalonate/isoprenoids dysregulation. Although mevalonate is upstream of CHOL, the mevalonate pathway has other branches and metabolites that may not be directly related to “CHOL biosynthesis.” In this setting, the manuscript title is misleading.

There does not seem any differential “CHOL pathway” dysregulation between stem cell, DCIS, and invasive carcinoma - they all have upregulated mevalonate pathway according to the article. Unless there are characteristic distinctions, the author’s claim that the CHOL pathway affects breast cancer stem cell, initiation, AND progression is overreaching – perhaps CHOL dysregulation is an early step and persists during tumorigenesis.

The section reviewing therapeutics targeting CHOL synthesis pathway could be more in-depth with appropriate references of specific clinical trials and epidemiological findings. As these therapeutics have practical implications in breast cancer treatment, expanding this section would be impactful.

This reviewer recommends revision.

Author Response

I thank the Reviewer for the comments. In particular,  I am pleased that the Reviewer has understood and appreciated the effort to combine the scientific rigor and the educational need to simplify a complex topic like this.

Under this perspective, it must be read the apparent misleading use of the terms "CHOL synthesis" and "mevalonate pathway" and the simplification of the interaction between CHOL biosynthesis and the Hippo signaling pathway.

The biosynthetic process leading to cholesterol is very complex, and the mevalonate cascade is only a portion of it; undoubtedly crucial, but still a portion. That is the reason why the title of the review refers to the biosynthetic process as a whole and not to the mevalonate pathway.

As for the interaction between CHOL biosynthesis and the Hippo signaling pathway, the apparent oversimplification was intentional, considering that most readers are unfamiliar with the Hippo signaling pathway and its complex interrelations with other metabolic pathways. As intentional was the oversimplification of Figure 2, which lacks the various molecules that sequester YAP in the cytoplasm and trigger its degradation, and the molecules that transduce the external signaling due to cell-cell contact, thus activating the Hippo cascade. 

As regards the section reviewing therapeutics targeting the CHOL synthesis pathway, I don't understand what specific clinical trials and epidemiological findings the Reviewer referred to, apart from the studies already cited in the review and supported by 15 references (116 - 131). 

Reviewer 2 Report

Comments and Suggestions for Authors

In this review the Author summarizes the up-to-date studies on the effect of increased cholesterol biosynthesis on breast cancer progression. My overall impression on the paper is high. The review is written in an elegant way and has a very high value, both in a scientific and educational sense. I wish every manuscript would be written in this way. The manuscript could be published almost without any changes. However, I have found a few minor flaws, mostly of editorial nature, that should be corrected before the publication:

- page 5, lines 171-173: there is an unnecessary repetition in the following sentence: “However, observational studies provided inconclusive and sometimes contradictory results on the relationship between circulating CHOL and breast cancer risk have provided inconclusive or contradictory results.” – it has to be changed to: “However, observational studies on the relationship between circulating CHOL and breast cancer risk have provided inconclusive or sometimes contradictory results.”;

- page 8, line 298: split one word “genecodes” into two separate ones “gene codes”.

Comments on the Quality of English Language

The English Language quality is fine. However, I have found one issue:

- page 4, line 145: the meaning of the following sentence is not clear: “Steroid hormones are, likely, the most CHOL derivatives.” I think that it would be better to write it in the following way: “Steroid hormones are, likely, primarily the CHOL derivatives.”.

Author Response

I thank the Reviewer for the kind words. I wish every manuscript revision was as positive as this one! I am pleased that the Reviewer has understood and appreciated the effort to combine the scientific rigor and the educational need to simplify a complex topic like this.

Additionally, I thank the Reviewer for the suggested changes, which will undoubtedly improve the quality of the article. Accordingly, I have removed the repetition on page 5, lines 171-173, and have replaced the old sentence with the one proposed by the Reviewer. Furthermore, as suggested, I have replaced the sentence on page 4, line 145. 

Round 2

Reviewer 1 Report

Comments and Suggestions for Authors

I didn’t see any edits in the v2 manuscript or maybe the edits were not readily visible.

Again, the author brings an interesting and important perspective on cholesterol as it relates to breast cancer. In general, it reads well. However, there are some generalizations and oversimplifications that are misleading to the research, clinical, and patient communities. These could be remedied by minor revisions as suggested below:

1.     The author could consider revising the title to cholesterol biosynthesis “pathway” rather than the biosynthesis of cholesterol since the review is not focused on cholesterol (eg cholesterol overproduction, underproduction, metabolites, etc).

2.     If the oversimplification of the Hippo/CHOL pathway interaction is intentional according to the author, it should be stated as such in the text(s). However, it’s not clear what aspects of oversimplification fit here. Prenylation facilitates activation of all small GTPases (e.g. Ras, Rho, Rab, Arf, etc) and small GTPases function via diverse mechanisms (e.g MAPK, actin etc). Thus, the cholesterol pathway (mevalonate pathway to be more appropriate) affecting geranylgeranylation of Rho, thereby specifically inhibiting the Hippo pathway and activating YAP/TAZ, is overreaching. As is written, it gives a false impression that mevalonate pathway only affects Hippo signaling. I would suggests removing this section or elaborating on the impact of CHOL pathway on prenylation as it relates to breast cancer.

3.     As for the therapeutics section, oversimplification of each drug is grossly misleading. For example, numerous studies including meta-analyses have found that no association between statins and breast cancer. There was one study (Danish study) if I remember correctly showed a specific type of statin, simvastatin, was associated with some efficacy. The authors should at least acknowledge or discuss these as it relates to the HMG-CoA inhibition vs simvastatin effect.

4.     The same caution can be applied to bisphosphonates – it’s a specific type of bisphosphonate that showed inhibition of farnesyl-PP synthetase. Inhibition of bone met by bisphosphonate is via a completely different mechanism (osteoclast inhibition). Thus, it should be clearly stated that it was a specific type of bisphosphonate, not all bisphosphonates.

5.     The author should at minimum directly acknowledge the pleiotropic effects of these agents and “tone down” the assertions in the text.

6.     Lastly, there are FPP or GGPP analogs as well as farnesyl or geranylgeranyl transferase inhibitors. I thought there were studies of these molecules in breast cancer. While these would add to the impact of the manuscript, this topic may warrant another review.

Author Response

Rebuttal To the Academic Editor

Dear Academic Editor,
I have received the second Review Report from Reviewer 1, which essentially reiterates the comments made in the first round. I have already explained my point of view about the feasibility of the review in my previous replies and will not be providing any further replies.

However, I would like to explain my disagreement with the Reviewer's comments, particularly regarding the "generalizations and oversimplifications" deemed "misleading to the research, clinical, and patient communities".
As I previously explained, the oversimplification of the Hippo/cholesterol relationship was intentional to make the review more accessible for readers who are not familiar with the subject matter. Indeed, the review focuses on the impact of the relationship between cholesterol production and the Hippo signaling pathway on breast tumorigenesis.  
Besides, as for the therapeutic approach to control the various steps of cholesterol biosynthesis, this topic was included in the last section titled "Conclusions and perspective" only to provide the reader with preliminary information about the different possibilities, till now explored, to control it. In fact, as correctly suggested by the Reviewer, this topic will deserve a specific review.  
Finally, consider the comment of Reviewer 2: "The review is written in an elegant way and has a very high value, both in a scientific and educational sense. I wish every manuscript would be written in this way. The manuscript could be published almost without any changes", I believe Reviewer 1's repeated criticisms are more provocative than constructive.
However, I will accept any of your final decisions, including the manuscript rejection.
Sincerely yours,
Danila Coradini